# Uterine cervical neoplasms mass screening at the University Hospital Centre of Libreville, Gabon: Associated factors with precancerous and cancerous lesions

Sylvain Honore Woromogo[1]*, Nathalie Ambounda Ledaga[1,2], Felicite Emma Yagata-Moussa[3], Astride Smeige Mihindou[2]

1 InterState Centre for Higher Public Health Education in Central Africa (CIESPAC), Brazzaville, Congo, 2 Gynecology and Obstetric Service, University Hospital Centre of Libreville, Libreville, Gabon, 3 Faculty of Health Sciences, University of Bangui, Bangui, Central African Republic

* woromogos@gmail.com

## Abstract

The objectives of this study were to identify the associated factors with cancerous and precancerous lesions of cervix. In Africa, the incidence of uterine cervical neoplasms varies from one region to another, where most women with uterine cervical neoplasms are seen at an advanced stage. For this reason, uterine cervical neoplasms mass screening reduces the incidence and mortality due to this disease, similar to what is being done in Europe. A cross-sectional analytical study was conducted. Socio-demographic characteristics, gynaecological-obstetrical history, risk factors, data from visual inspection with acetic acid and visual inspection with Lugol, colposcopy impressions and results of cytological analysis were performed. A simple and multiple regression were performed to establish a statistically significant difference between certain factors and the presence of precancerous or cancerous lesions of uterine cervical. In this study, of 63 women diagnosed histologically, 43 had precancerous lesions and 20 had cancerous lesions. we found that being older than 35, having the first intercourse before 18, having an antecedent of STI, being a widow and using of tobacco were risk factors associated with precancerous lesions (p = 0.013 with OR = 3.44 (1.22–9.73), p = 0.009 with OR = 4.07 (1.69–13.08), p < 0.001 with OR = 3.80 (1.94–7.47), p < 0.001 with OR = 9.77 (3.87–24.70) and p < 0.001 with OR = 5.47 (2.60–11.52)) respectively. Only being older than 45, being a widow and using tobacco were risk factors associated with cancerous lesions (p = 0.021 with OR = 2.01 (1.58–3.56), p = 0.02 with OR = 2.96 (2.10–3.87), p = 0.041 with OR = 1.98 (1.46–2.44)) respectively. Among participants diagnosed with uterine cervical neoplasms, there was a significant association with the STI, marital status and smoking. Despite the integration of the detection of precancerous uterine cervical neoplasms lesions into health facilities in Gabon, uterine cervical neoplasms ranks second among women's cancers in terms of incidence and first in terms of mortality.

**Data Availability Statement:** All relevant data are within the manuscript.

**Funding:** The authors received no specific funding for this work.

**Competing interests:** The authors have declared that no competing interests exist.

## Introduction

Uterine Cervical Neoplasm is caused by untreated and undiagnosed precancerous lesions that are not diagnosed in time. According to GLOBOCAN 2018, the number of new cases of uterine cervical neoplasms is estimated at 569,847 representing 3.2% of all 36 cancers with 569,847 deaths representing 3.2% of all cancer deaths [1]. Also, standardized incidence rates per 100,000 and standardized mortality rates per 100,000 were higher in developing countries than in developed countries for uterine cervical neoplasms with eight versus five and four versus two respectively [2]. Each year more than 500 women are diagnosed with uterine cervical neoplasm and the disease causes more than 300,000 deaths worldwide [3]. In Africa, the incidence of uterine cervical neoplasms varies from one region to another, particularly in developing countries where most women with uterine cervical neoplasms are seen at an advanced and often incurable stage or are only suitable for palliation [4, 5]. For this reason, uterine cervical neoplasms mass screening reduces the incidence and mortality due to this disease [6], similar to what is being done in Europe [7]. The World Health Organization (WHO) and other agencies have proposed guidelines for mass screening [6, 8, 9] and recommendations for abnormal cervical smears [10–12]. In Gabon, early detection of uterine cervical neoplasms has been integrated into health facilities since 2014. But little data is available on the prevalence of precancerous lesions in diagnostic centres. The objective of this study is to determine the prevalence of precancerous cervical lesions in women screened at University Hospital Centre of Libreville (UHCL) in 2018 and to identify the factors associated with these lesions.

## Materials and methods

### Type of study

This was a cross-sectional analytical study conducted during the period from January 1, 2018 to December 31, 2018. Women who came for screening and agreed to participate in the study were consecutively included.

### Study population

The study population consisted of women aged 25 to up 65 years old who were screened for uterine cervical neoplasms in Libreville. Sensitization on uterine cervical neoplasms screening was done by radio advertisements and with the help of community health workers before and during the screening period. Participants were recruited from the population of Libreville without regard to HIV status. Women were excluded from the study if they were more than 20 weeks pregnant; less than 12 weeks post-partum; had a previous history of treatment of cancerous lesions; had a known allergy to acetic acid; or if they had undergone total hysterectomy.

### Data collection procedures

Screening was done by medical doctors, nurses, and midwives with 3 years of experience in maternity service who had been trained for 2 weeks in theory and practice in uterine cervical neoplasms mass screening. The data were collected from a questionnaire and results forms for cytopathological anatomy analysis. This screening consisted in carrying out successively (See Fig 1):

A speculum examination during which the Visual inspection after application of acetic acid (VIA) and lugol's iodine (VIL) tests are performed for screening. When there is an anomaly in the VIA and VIL (positive VIA, VIL test), a colposcope examination is systematically performed to clarify the lesion and guide the biopsy. The biopsy fragments are immediately attached to 10% formaldehyde and sent to the laboratory for histological examination.

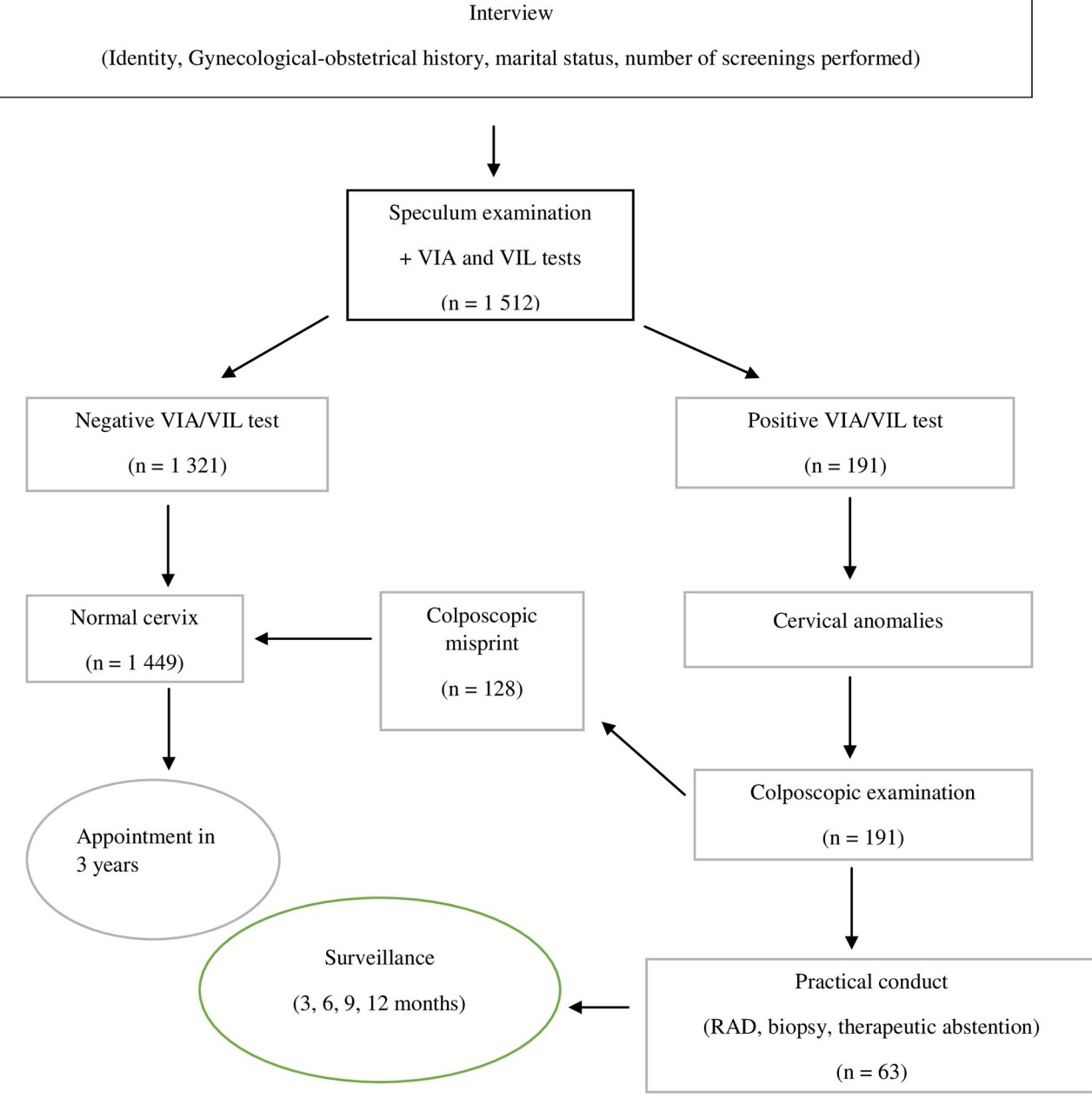

**Fig 1. Patient inclusion and screening steps.**

The histological results of the piece are brought back to the centre by the participant herself in order to develop a therapeutic plan and follow-up with a well-defined monitoring rhythm and consequent intervention. All women were treated according to the protocol recommended by National Agency for Health Accreditation and Evaluation [11, 13].

## Variables

The following variables were studied: socio-demographic characteristics, gynaecological-obstetrical history, risk factors, data from visual inspection with acetic acid and visual

inspection with Lugol, colposcopy impressions and results of cytological and histological analysis when a biopsy was performed. Two dependent variables of interest were chosen. Invasive uterine cervical neoplasm was defined as a binary variable (yes/no) with an outcome of diagnosis of invasive uterine cervical neoplasm. Invasive uterine cervical neoplasm lesions were diagnosed clinically if lesions suggestive of invasive cancer were found on the uterine cervical. All cases were confirmed by colposcopy and biopsy. Cervical pre-cancer was defined as a binary variable. Cervical pre-cancer lesions were diagnosed clinically after application of acetic acid [14–16]. Independent variables included socio-demographic variables: these included age, sex, educational level, marital status, tobacco, age of first intercourse, STIs, hormonal status, gesture, parity.

## Statistical analysis

Text and table entry was done on Word and Excel 2016 software and data analysis was done with EpiInfo 7.2 softwares. The socio-demographic and gynaecological characteristics as well as the results of various tests were obtained by simple frequencies. Summary statistics such as mean and standard deviation were calculated for continuous variables. Bivariate methods were used to show the relationship between variables; contingency tables were used to describe relationship between categorical variables. Tabular methods of describing the relationship between two nominal variables by finding proportions were also employed. Multivariate analysis was used to determine the socio-demographic, behavioral and clinical factors associated with uterine cervical neoplasm and pre-cancer lesions. Binary and multivariate logistic regressions were used to show the association of any cervical cancerous lesion with different factors To facilitate regressions, most variables were made into binary variables (age group, marital status, education level, social economic status, number of pregnancies, number of children born, age of the first intercourse) to facilitate analysis of the association of uterine cervical neoplasm and precancer with different factors.

The Chi-square and Wald tests were used as well as the odds ratio with their 95% confidence interval at the 5% threshold. Multiple logistic regression was used to establish the relationship between socio demographic and gynecological characteristics and precancerous or cancerous lesions. Multivariable logistic regression analyses with backward elimination stepwise selection with $p < 0.20$ were used to identify baseline explication that predicts precancerours or cancerous lesions.

## Ethical considerations

The study was conducted in accordance with the Good Clinical Practices (GCP) guidelines and the regulations of the Ministry of Public Health and Population. The study was approved by the Ethics Committee of the University of Health Sciences, Gabon with the protocol number for ethical approval is 007/USS/18. All women admitted to the programme were admitted after obtaining their written informed consent.

## Results

The average age of the women screened was 39.8 ± (SD = 11.4) years with extremes of 25 and 83 years. Women aged 25–34 were the most represented, in the order of 40.5% of the sample. There were 625 women with a history of sexually transmitted infections, 41.3%. Of these, 534 had Chlamydia infection, 35 had HIV and 56 had gonorrhoea and syphilis. Smoking women accounted for 5.8% of the participants. Among the participants screened, 67.8% did not use a contraceptive method. The other characteristics of women are presented in Table 1.

**Table 1. Sociodemographic and medical characteristics of participants.**

| Variables | | Number | Percentage |
|---|---|---|---|
| Age (years) | | | |
| | 25–34 | 613 | 40.5 |
| | 35–44 | 427 | 28.2 |
| | 45–54 | 292 | 19.3 |
| | 55–64 | 141 | 9.3 |
| | 65 + | 39 | 2.6 |
| Marital status | | | |
| | Married | 425 | 28.1 |
| | Concubine/single | 1024 | 67.7 |
| | Widow | 63 | 4.2 |
| Hormonal status | | | |
| | Postmenopausal | 297 | 19.6 |
| | Premenopausal | 1215 | 80.4 |
| Current contraceptive use | | | |
| | Yes | 487 | 32.2 |
| | No | 1025 | 67.8 |
| Gesture | | | |
| | Nulligest | 155 | 10.3 |
| | Primigest | 154 | 10.2 |
| | Multigest | 1203 | 79.5 |
| Parity | | | |
| | Nulliparous | 206 | 13.6 |
| | Primiparous | 308 | 20.4 |
| | Multiparous | 998 | 66.0 |
| First intercourse (years) | | | |
| | < 12 | 41 | 2.7 |
| | 12–17 | 802 | 53.1 |
| | 18–29 | 669 | 44.2 |
| STIs | | | |
| | Yes | 625 | 41.3 |
| | No | 887 | 58.7 |
| Tobacco | | | |
| | Yes | 87 | 5.8 |
| | No | 1425 | 94.2 |

## Visual inspection and colposcopy results

In this study, 12.0% of women had a pathological cervix on unprepared examination. Women who tested positive for VIA and VIL accounted for 5.9% and 12.6% respectively. Colposcopy was only for cases with positive VIL, 191 tests or 12.6% (191/1512). With colposcopy, 46.1% of women had atypical grade1 transformations, 49.7% grade 2 and 4.2% had both atypical transformations (Table 2).

## Histological results

In this study, of 63 women diagnosed histologically, 43 had precancerous lesions with 23 CIN I or 36.5%; 11 CIN II or 17.5% and 9 CIN III or 14.3%. Squamous cell cancers accounted for 20.6% and adenocarcinomas 3.2% (Table 3).

**Table 2. Distribution of women according to different types of tests.**

| Visual inspection | N | | | Number | Percentage |
|---|---|---|---|---|---|
| Unprepared test | 1512 | | | | |
| | | | Normal | 1327 | 88.0 |
| | | | Pathological | 185 | 12.0 |
| VIA test | 1512 | | | | |
| | | | Negative | 1423 | 94.1 |
| | | | Positive | 89 | 5.9 |
| VIL test | 1512 | | | | |
| | | | Negative | 1321 | 87.4 |
| | | | Positive | 191 | 12.6 |
| Colposcopy* | 191 | | | | |
| | | | ATG 1 | 88 | 46.1 |
| | | | ATG 2 | 95 | 49.7 |
| | | | ATG 1 & ATG 2 | 08 | 4.2 |

* Positive cases only from VIL test

## Risk factors of precancerous and cancerous lesions

Tables 4 and 5 show the characteristics that were risk factors for precancerous and cancerous lesions using bivariate and multivariable analysis. Using binary logistic regression to assess the relationship between cervical precancerous lesions and the independent variables collected, we found that being older than 35, having the first intercourse before 18, having an antecedent of STI, being a widow and use of tobacco were risk factors associated with precancerous lesions (p = 0.013 with OR = 3.44 (1.22–9.73), p = 0.009 with OR = 4.07 (1.69–13.08), p < 0.001 with OR = 3.80 (1.94–7.47), p < 0.001 with OR = 9.77 (3.87–24.70) and p < 0.001 with OR = 5.47 (2.60–11.52)) respectively. With multivariate regression, the risk factors of developing any precancerous lesion were the same but the OR decreased with increased parity (p = 0.04 with OR = 1.44 (1.09–2.21), p = 0.04 with OR = 2.01 (1.99–2.61), p = 0.006 with OR = 1.97 (1.68–2.89), p = 0.021 with OR = 1.99 (1.30–2.37) and p = 0.02 with OR = 2.29 (1.14–2.99)) respectively. The total number of pregnancies and parity were not associated with precancerous lesions and cervical cancer. But after adjustement, only being older than 45, being a widow and using tobacco were risk factors associated with cancerous lesions (p = 0.021 with OR = 2.01 (1.58–3.56), p = 0.02 with OR = 2.96 (2.10–3.87), p = 0.041 with OR = 1.98 (1.46–2.44)) respectively.

**Table 3. Results from histological examinations for screened women.**

| | Histology | Number | Percentage |
|---|---|---|---|
| Precancerous lesions | | | |
| | CIN 1 | 23 | 36.5 |
| | CIN 2 | 11 | 17.5 |
| | CIN 3 | 9 | 14.3 |
| Cervical cancers | | | |
| | Squamous cell cancers | 13 | 20.6 |
| | Adenocarcinomas | 07 | 11.1 |
| Total | | 63 | 100.0 |

**Table 4. Risk factors of precancerous lesions.**

| Risk factors | Histology | | | p | AOR* (95% CI) | p |
|---|---|---|---|---|---|---|
| | Injury | No injury | OR (95% CI) | | | |
| Age (years) | | | | | | |
| 25–34 | 05 | 608 | | 1.0 | 1.0 | - |
| 35–44 | 25 | 402 | 7.56 (2.87–19.91) | < 0.001 | 2.81 (2.09–3.11) | 0.05 |
| 45 + | 13 | 459 | 3.44 (1.22–9.73) | 0.013 | 1.44 (1.09–2.21) | 0.04 |
| Age on first intercourse | | | | | | |
| < 12 | 05 | 36 | 6.06 (2.08–17.59) | 0.004 | 1.66 (1.11–3.36) | 0.03 |
| 12–17 | 23 | 779 | 4.07 (1.69–13.08) | 0.009 | 2.01 (1.99–2.61) | 0.04 |
| 18–29 | 15 | 654 | 1.0 | - | 1.0 | |
| STIs | | | | | | |
| Yes | 31 | 594 | 3.80 (1.94–7.47) | < 0.001 | 1.97 (1.68–2.89) | 0.006 |
| No | 12 | 875 | 1.0 | - | 1.0 | - |
| Marital status | | | | | | |
| Married | 09 | 416 | 1.0 | - | | |
| Concubine/Single | 23 | 1021 | 1.04 (0.48–2.27) | 0.55 | | |
| Widow | 11 | 52 | 9.77 (3.87–24.70) | < 0.001 | 1.99 (1.30–2.37) | 0.021 |
| Tobacco | | | | | | |
| Yes | 10 | 77 | 5.47 (2.60–11.52) | < 0.001 | 2.29 (1.14–2.99) | 0.02 |
| No | 33 | 1392 | 1.0 | | 1.0 | |

*: Adjusted Odds ratio.

Adjustment on gesture, parity and use of oral contraceptive

**Table 5. Risk factors for cancerous lesions.**

| Risk factors | Histology | | | p | AOR* (95% CI) | p |
|---|---|---|---|---|---|---|
| | Injury | No injury | OR (95% CI) | | | |
| Age (years) | | | | | | |
| 25–34 | 3 | 610 | 1.0 | | 1.0 | |
| 35–44 | 6 | 421 | 2.89 (0.72–11.65) | 0.11 | - | |
| 45 + | 11 | 461 | 4.85 (1.35–17.49) | 0.008 | 2.01 (1.58–3.56) | 0.021 |
| Age on first intercourse | | | | | | |
| < 12 | 4 | 37 | 10.22 (2.86–36.49) | 0.0023 | 1.88 (0.97–2.01) | 0.087 |
| 12–17 | 9 | 793 | 1.17 (0.40–2.89) | 0.547 | | |
| 18–29 | 7 | 662 | 1.0 | - | | |
| STIs | | | | | | |
| Yes | 14 | 611 | 3.36 (1.29–8.80) | 0.008 | 1.97 (1.68–2.89) | 0.050 |
| No | 06 | 881 | 1.0 | - | 1.0 | - |
| Marital status | | | | | | |
| Married | 04 | 421 | 1.0 | - | | |
| Concubine/Single | 07 | 1017 | 0.72 (0.21–2.49) | 0.410 | | |
| Widow | 09 | 63 | 15.03 (4.49–50.28) | < 0.001 | 2.96 (2.10–3.87) | 0.025 |
| Tobacco | | | | | | |
| Yes | 05 | 82 | 5.73 (2.03–16.16) | 0.004 | 1.98 (1.46–2.44) | 0.041 |
| No | 15 | 1410 | 1.0 | | 1.0 | |

*: Adjusted Odds ratio.

Adjustment on gesture, parity and use of oral contraceptive

## Discussion

This study provided an update on the prevalence of precancerous cervical lesions in the obstetrics and gynaecology department. The target population was sexually active women 25 years of age and older. Several countries have succeeded in preventing 91% of all invasive cervical cancers by implementing generalized cytological screening [17]. This study is a first-time study in Gabon. Considering that early detection of uterine cervical neoplasms has been integrated into health facilities since 2014, we aimed to identify the factors associated with cervical cancer in the Gabonese context and compare them with those described in the literature [1, 14–16]

### Socio-demographic and gyneco-obstetrical characteristics

In this study, the mean age was 39.8 ±11.4 years with extremes ranging from 25 to 83 years. Some authors have worked on the same subject as us. For these authors, the average age ranged from 33–40 years [7, 9, 10, 17–20] and was close to ours. The age at which screening begins is the subject of much debate among uterine cervical neoplasm screening professionals. At the global level, there is no consensus on the age at which screening should begin. At 21 years of age (or one year after the first reports) in the United States and Switzerland [2], 25 years in England, France and according to WHO [3, 4]. Uterine cervical neoplasm is caused by an oncogenic sexually transmitted HPV infection. Therefore, any woman who has already had sexual intercourse should be regularly screened. The majority of the study population were single and concubine women (67.7%), however 28.1% were married. These results are similar to those found by MPIGA et al [18]. The age at which screening begins in Gabon is 25 years old. It was found that 80.4% of participants were premenopausal. Paradoxically, 67.8% of women were not using contraceptives. This translates into a small percentage of nulliparous people. In Gabon, women have the perception that contraception makes them sterile.

### Screening results

In this series, 91 women had a pathological cervix, or 2%. The VIA and VIL tests were positive at 5.9% and 12.6% respectively. These results vary in some authors from 1.9% to 10.8% for VIA [10, 18, 21–23] and from 4.8% to 5.8% for VIL [13, 18]. This difference is explained by the fact that women have been sensitized with an important message: uterine cervical neoplasm screening reduces the risk of disease and mortality. The cytohistological analysis determined the prevalence of precancerous lesions at 2.8%. This prevalence seems high because of the large number of women screened, compared to some authors [23]. The screening programme should focus on raising awareness among women, which is generally very successful and has shown that countries that have implemented cervical cancer screening reduce the burden of this disease [24, 25].

### Risk factors

We found that 41.3% of the women screened reported having contracted a STI. In our data base most of them contracted chlamydia infection and gonococcal disease. This high percentage of chlamydia infection is of concern; Bahmanyar et al in 2012 has established that history of sexually transmitted infections (STIs) is significantly associated with a risk of cervical injury [26]. Unprotected sexual activity is associated with an increased risk of contracting STIs, leading to cervical inflammation which can promote the development of cervical dysplasia [27]. In this study 5.8% of the participants were smokers. Smoking seems to be strongly associated with the development of precancerous cervical lesions and cancer. Smoking is one of the most commonly identified environmental co-factors that can affect the risk of uterine cervical

neoplasm. The majority of women screened, 53.1% had their first sexual intercourse between 12–17 years of age. This result marks the precocity of sexual intercourse, which is a factor that favours precancerous lesions of the cervix [18, 20, 28]. It is important to continue with raising awareness among women about screening. Compared to married women, widows are more prone to develop precancerous and cancerous cervical lesions. We could not explore the reasons for widowhood, but many of these widows are on antiretroviral treatment. These results confirm the importance of organizing uterine cervical neoplasm mass screening in Gabon. The countries that have implemented this screening programme have made a significant contribution to reducing the burden of this disease [25].

This study had several limitations. As the study was a convenience sample of women presenting for care from only Libreville in Gabon, the prevalence results may not be generalizable to the whole of Gabon or to the Central African female population, where HIV infection rates are quite varied. There is also a possibility of selection bias, as women who participated in the screening program may have been more likely to have current symptoms. The main purpose of this study was to serve as a baseline for further studies and to help for planning the uterine cervical neoplasms screening in Gabon; future studies using randomized sampling techniques may give a more complete estimation of the national prevalence. Lastly, there were limited variables available for analysis in the medical records, limiting the number of risk factors and confounders that we were able to study, for exemple the antecedent of cancer family.

## Conclusion

In this study, the participants in the uterine cervical neoplasm mass screening at the UHCL in 2018 had an average age of 39 years, they were mostly single, premenopausal, multi-gestured. Early sexual intercourse and smokers were found. Among participants diagnosed with uterine cervical neoplasm, there was a significant association with age, STI, marital status and smoking. Despite the integration of the detection of precancerous cervical cancer lesions into health facilities in Gabon, uterine cervical neoplasm ranks second among women's cancers in terms of incidence and first in terms of mortality. These results help to further strengthen the fight against uterine cervical neoplasm, provide clinicians with the characteristics of women with uterine cervical neoplasms and allow further research to identify the factors that contribute to it. Thus, the detection and management of precancerous cervical lesions could reduce the mortality rate from uterine cervical neoplasm in the long term.

## Acknowledgments

The authors would like to thank the Ministry of Health of Gabon.

## Author Contributions

**Conceptualization:** Nathalie Ambounda Ledaga.

**Investigation:** Nathalie Ambounda Ledaga, Astride Smeige Mihindou.

**Methodology:** Sylvain Honore Woromogo, Felicite Emma Yagata-Moussa, Astride Smeige Mihindou.

**Validation:** Sylvain Honore Woromogo.

**Visualization:** Felicite Emma Yagata-Moussa.

**Writing – original draft:** Sylvain Honore Woromogo, Felicite Emma Yagata-Moussa, Astride Smeige Mihindou.

**Writing – review & editing:** Felicite Emma Yagata-Moussa.

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
