## [Decision Letter · Decision Letter 0]

6 Jan 2021

PONE-D-20-32144

Cervical cancer screening at the University Hospital Centre of Libreville, Gabon: associated factors with precancerous and cancerous lesions

PLOS ONE

Dear Dr. WOROMOGO,

Thank you for submitting your manuscript to PLOS ONE. After careful consideration, we feel that it has merit but does not fully meet PLOS ONE’s publication criteria as it currently stands. Therefore, we invite you to submit a revised version of the manuscript that addresses the points raised during the review process. Details you will find in the reviewers´reports below.

We look forward to receiving your revised manuscript.

Kind regards,

Konradin Metze

Academic Editor

PLOS ONE

Journal Requirements:

2. For more information on PLOS ONE's expectations for statistical reporting, please see https://journals.plos.org/plosone/s/submission-guidelines.#loc-statistical-reporting

Please update your Methods and Results sections accordingly.

'The funders had no role in study design, data collection and analysis, decision to publish, or preparation of the manuscript.'

Reviewers' comments:

Reviewer's Responses to Questions

**Comments to the Author**

1. Is the manuscript technically sound, and do the data support the conclusions?

Reviewer #1: Yes

Reviewer #2: Partly

Reviewer #3: Yes

2. Has the statistical analysis been performed appropriately and rigorously? 

Reviewer #1: Yes

Reviewer #2: Yes

Reviewer #3: I Don't Know

3. Have the authors made all data underlying the findings in their manuscript fully available?

Reviewer #1: Yes

Reviewer #2: Yes

Reviewer #3: Yes

4. Is the manuscript presented in an intelligible fashion and written in standard English?

Reviewer #1: Yes

Reviewer #2: Yes

Reviewer #3: Yes

5. Review Comments to the Author

Reviewer #1: I have marked several things in the manuscript that need to be addressed. While I feel that this a well written manuscript, the things that I have identified within the attachment need change or clarification. In addition, please explain why the risk factors are not the same for Tables 4 & 5. Please discuss why widows have such high risk. It seems to me that the table support precancerous and cancerous lesions having nearly identical risk factors. I think the authors should speak to the question of the power of the study. It is still acceptable for publication if it is underpowered, but for the sake of the reader this should be included. Lastly, I recommend a final paragraph titled "Why this is important" which should be entirely textual.

Reviewer #2: The objectives of the study were to identify factors associated with cancerous and precancerous cervical lesions in Gabon in women screened at the University Hospital Center of Libreville (UHCL) to identify factors associated with these lesions.

The question was not described, as the study design and the small sample size were not appropriate for more comprehensive analyzes. Most of the precancerous lesions were CIN 1, which clinically is not a true precancerous lesion. Finally, the conclusions were not supported by the findins.

In summary, the objective of the study can be considered interesting for a better characterization of some variables about cervical cancer in this specific region, but the general interest is limited in order to meet the requirements of the journal.

Reviewer #3: The article, it deals with a current topic and of great importance for the woman's health, although it is already well clarified in the literature. The article is simple, but provides an overview of the current situation in the country of the authors. It uses a simple statistical technique (EpiInfo program) but sufficient.

I made considerations that I consider important for the publication of the article, below:

Please review the summary and do not include references in this section; The keywords must be extracted from the Mesh controlled vocabulary and not be free and random words; Please inform the number of the ethical approval protocol for this study; I suggest inserting in the discussion session what the research differs from its conventional local treatment; I suggest inserting a patient inclusion flowchart; insert in the “variables” section the citations and bibliographic references to classify the types of cancers used as a cutoff point in the study, in the “conclusion” section please highlight how your findings contribute to clinical practice and future research.

Success to authors,

6. PLOS authors have the option to publish the peer review history of their article (what does this mean?). If published, this will include your full peer review and any attached files.

Reviewer #1: No

Reviewer #2: No

Reviewer #3: No

---

## [Author Response · Author response to Decision Letter 0]

12 Feb 2021

Response to Reviewers

Reviewer #1

Thank you very much for the importance given to our manuscript. We find your comments very pertinent. We have taken these remarks into consideration and corrections have been made in the text.

The number of cases of precancerous lesions is higher than that of cancerous lesions for each risk factor studied. This explains why the risk factors are not the same for Tables 4 and 5. We also know that precancerous lesions, if diagnosed in early detection, can be treated and cured.

In our database, 8 out of 9 widows with cancerous lesions are over 45 years old. Similarly, all widows with precancerous lesions are over 45 years old. We were therefore unfortunately unable to investigate further to find out the reasons of widowhood.

The question of the power of the study is raised as a limitation of the study in the discussion.

Reviewer #2

Thank you very much for the importance given to our manuscript. We find your comments very pertinent. We have taken these remarks into consideration and corrections have been made in the text. Indeed, the objectives of the study were to identify factors associated with cancerous and precancerous cervical lesions. And factors such as age, sexually transmitted infections, smoking and other factors have been incriminated. Nevertheless, we have just included in the discussion the power of the study as a limitation to our study.

Reviewer #3

Thank you very much for the importance given to our manuscript. We find your comments very pertinent. We have taken these remarks into consideration and corrections have been made in the text.

We have reviewed the abstract and the references have been removed. The keywords have been revised and extracted from the Mesh controlled vocabulary. The number of the ethical approval protocol for this study is included in the text (methods). 

We have inserted in the discussion what the research differs from its conventional local treatment. In the “conclusion” section we highlighted how our findings contribute to clinical practice and future research.

We have inserted figure 1 for patient inclusion and screening steps. If you think it is not correct, it could be removed from the text.

---

## [Decision Letter · Decision Letter 1]

15 Mar 2021

PONE-D-20-32144R1

Cervical cancer screening at the University Hospital Centre of Libreville, Gabon: associated factors with precancerous and cancerous lesions

PLOS ONE

Dear Dr. WOROMOGO,

Thank you for submitting your manuscript to PLOS ONE. After careful consideration, we feel that it has merit but does not fully meet PLOS ONE’s publication criteria as it currently stands. Therefore, we invite you to submit a revised version of the manuscript that addresses the points raised during the review process. 

Please read the reviewers´  comments  copied below. 

We look forward to receiving your revised manuscript.

Kind regards,

Konradin Metze

Academic Editor

PLOS ONE

Journal Requirements:

Reviewers' comments:

Reviewer's Responses to Questions

**Comments to the Author**

1. If the authors have adequately addressed your comments raised in a previous round of review and you feel that this manuscript is now acceptable for publication, you may indicate that here to bypass the “Comments to the Author” section, enter your conflict of interest statement in the “Confidential to Editor” section, and submit your "Accept" recommendation.

Reviewer #1: All comments have been addressed

Reviewer #3: All comments have been addressed

2. Is the manuscript technically sound, and do the data support the conclusions?

Reviewer #1: Partly

Reviewer #3: Yes

3. Has the statistical analysis been performed appropriately and rigorously? 

Reviewer #1: Yes

Reviewer #3: I Don't Know

4. Have the authors made all data underlying the findings in their manuscript fully available?

Reviewer #1: Yes

Reviewer #3: Yes

5. Is the manuscript presented in an intelligible fashion and written in standard English?

Reviewer #1: Yes

Reviewer #3: Yes

6. Review Comments to the Author

Reviewer #1: Some reservation exists regarding the sample size / power estimate. The re-submission is greatly improved. I have highlighted a few places in the revision that require textual consideration.

Reviewer #3: Thank you again for the opportunity to review this work.

The following items still need attention:

Please check if "Screening" and "Neoplasia" are indexed to Mesh.

I suggest changing "Cervix cancer" to "Uterine Cervical Neoplasms".

The inclusion of the patient inclusion flowchart clarified the work process for the reader. I suggest including the number of eligible patients at each stage of the process.

Success to the authors!

7. PLOS authors have the option to publish the peer review history of their article (what does this mean?). If published, this will include your full peer review and any attached files.

Reviewer #1: No

Reviewer #3: No

---

## [Author Response · Author response to Decision Letter 1]

21 Apr 2021

Response to Reviewers

Journal requirements

We reviewed our reference list and found that it is complete and correct.

Reviewer #1

Thank you very much for the importance given to our manuscript. We find your comments very pertinent. We have taken these remarks into consideration and corrections have been made in the text.

As this was a mass screening campaign that went on for some time, we enrolled all eligible women who presented themselves

Reviewer #3

After checking, "Screening" is indexed as "Mass Screening" and "Neoplasia" is indexed as "Neoplasms" to MeSH. 

"Cervix cancer" changed to "Uterine Cervical Neoplasms" in the text.

The number of eligible patients at each stage of the process is included in the flowchart.

---

## [Decision Letter · Decision Letter 2]

17 May 2021

PONE-D-20-32144R2

Uterine cervical Neoplasms mass screening at the University Hospital Centre of Libreville, Gabon : associated factors with precancerous and cancerous lesions

PLOS ONE

Dear Dr. WOROMOGO,

Thank you for submitting your manuscript to PLOS ONE. After careful consideration, we feel that it has merit but does not fully meet PLOS ONE’s publication criteria as it currently stands. Therefore, we invite you to submit a revised version of the manuscript that addresses the points raised during the review process.

Unfortunately not all suggestions of the reviewers were followed. 

Please address adequately the former and actual comments of the reviewers. It is necessary to include  the protocol number of the ethical approvement in the material and methods section. 

We look forward to receiving your revised manuscript.

Kind regards,

Konradin Metze

Academic Editor

PLOS ONE

Journal Requirements:

Reviewers' comments:

Reviewer's Responses to Questions

**Comments to the Author**

1. If the authors have adequately addressed your comments raised in a previous round of review and you feel that this manuscript is now acceptable for publication, you may indicate that here to bypass the “Comments to the Author” section, enter your conflict of interest statement in the “Confidential to Editor” section, and submit your "Accept" recommendation.

Reviewer #1: All comments have been addressed

Reviewer #3: (No Response)

2. Is the manuscript technically sound, and do the data support the conclusions?

Reviewer #1: Yes

Reviewer #3: Partly

3. Has the statistical analysis been performed appropriately and rigorously? 

Reviewer #1: Yes

Reviewer #3: I Don't Know

4. Have the authors made all data underlying the findings in their manuscript fully available?

Reviewer #1: No

Reviewer #3: Yes

5. Is the manuscript presented in an intelligible fashion and written in standard English?

Reviewer #1: Yes

Reviewer #3: Yes

6. Review Comments to the Author

Reviewer #1: The authors have improved the original manuscript. It is submitted as a baseline study for their area and should be helpful to cervical cancer efforts moving forward.

Reviewer #3: Dear authors, hello

Previously, I made considerations that I consider important for the publication of the article and the vast majority was not attended to. My suggestions were only included in the summary, where the authors removed the references and the keywords were extracted from Mesh's controlled vocabulary and are no longer free and random words;

It was not corrected: insertion of the protocol number for the ethical approval of this study; I suggest inserting in the discussion session what the research differs from its conventional local treatment; I suggest inserting a patient inclusion flowchart; insert in the “variables” section the citations and bibliographic references to classify the types of cancer used as a cutoff point in the study, in the “conclusion” section highlight how your findings contribute to clinical practice and future research.

The above requests have not been met and there is no justification for this to occur.

Thanks for listening

7. PLOS authors have the option to publish the peer review history of their article (what does this mean?). If published, this will include your full peer review and any attached files.

Reviewer #1: No

Reviewer #3: No

---

## [Author Response · Author response to Decision Letter 2]

26 Jun 2021

Response to Reviewers

Journal requirements

We reviewed our reference list and found that it is complete and correct.

Reviewer #3

Thank you very much for your contribution for this study.

The protocol number for the ethical approval of this study is included; 

What the research differs from its conventional local treatment is discussed; 

Inserting a patient inclusion flowchart : it is ok

Inserting in the “variables” section the citations and bibliographic references to classify the types of cancer used as a cutoff point in the study : it is done

In the “conclusion” section highlight how your findings contribute to clinical practice and future research : it is done

---

## [Decision Letter · Decision Letter 3]

14 Jul 2021

Uterine cervical Neoplasms mass screening at the University Hospital Centre of Libreville, Gabon : associated factors with precancerous and cancerous lesions

PONE-D-20-32144R3

Dear Dr. WOROMOGO,

We’re pleased to inform you that your manuscript has been judged scientifically suitable for publication and will be formally accepted for publication once it meets all outstanding technical requirements.

Kind regards,

Konradin Metze

Academic Editor

PLOS ONE

Additional Editor Comments (optional):

Reviewers' comments:

Reviewer's Responses to Questions

**Comments to the Author**

1. If the authors have adequately addressed your comments raised in a previous round of review and you feel that this manuscript is now acceptable for publication, you may indicate that here to bypass the “Comments to the Author” section, enter your conflict of interest statement in the “Confidential to Editor” section, and submit your "Accept" recommendation.

Reviewer #3: All comments have been addressed

2. Is the manuscript technically sound, and do the data support the conclusions?

Reviewer #3: Partly

3. Has the statistical analysis been performed appropriately and rigorously? 

Reviewer #3: I Don't Know

4. Have the authors made all data underlying the findings in their manuscript fully available?

Reviewer #3: No

5. Is the manuscript presented in an intelligible fashion and written in standard English?

Reviewer #3: Yes

6. Review Comments to the Author

Reviewer #3: (No Response)

7. PLOS authors have the option to publish the peer review history of their article (what does this mean?). If published, this will include your full peer review and any attached files.

Reviewer #3: No

---

## [Editor Report · Acceptance letter]

16 Jul 2021

PONE-D-20-32144R3 

Uterine cervical Neoplasms mass screening at the University Hospital Centre of Libreville, Gabon : associated factors with precancerous and cancerous lesions 

Dear Dr. Woromogo:

I'm pleased to inform you that your manuscript has been deemed suitable for publication in PLOS ONE. Congratulations! Your manuscript is now with our production department. 

Kind regards, 

on behalf of

Prof. Konradin Metze 

Academic Editor

PLOS ONE